# Application of Modified Spent Mushroom Compost Biochar (SMCB/Fe) for Nitrate Removal from Aqueous Solution

**DOI:** 10.3390/toxics9110277

**Published:** 2021-10-21

**Authors:** Negisa Darajeh, Hossein Alizadeh, David W. M. Leung, Hamid Rashidi Nodeh, Shahabaldin Rezania, Hossein Farraji

**Affiliations:** 1School of Biological Sciences, University of Canterbury, Christchurch 8041, New Zealand; david.leung@canterbury.ac.nz; 2Bio-Protection Research Centre, Lincoln University, Lincoln 7647, New Zealand; Hossein.Alizadeh@lincoln.ac.nz; 3Food Technology and Agricultural Products Research Centre, Standard Research Institute, Karaj 3174734563, Iran; rnhamid2@gmail.com; 4Department of Environment and Energy, Sejong University, Seoul 05006, Korea; shahab.rezania@sejong.ac.kr; 5School of Physical and Chemical Sciences, University of Canterbury, Christchurch 8041, New Zealand; hossein.farraji@canterbury.ac.nz

**Keywords:** modified spent mushroom compost biochar (SMCB/Fe), nitrate removal, aqueous solution, adsorbent

## Abstract

The public is already aware that nitrate pollution caused by nutrient runoff from farms is harmful to aquatic life and human health, and there is an urgent need for a product/technology to solve this problem. A biochar adsorbent was synthesized and used to remove nitrate ions from aqueous media based on spent mushroom compost (SMC), pre-treated with iron (III) chloride hexahydrate and pyrolyzed at 600 °C. The surface properties and morphology of SMCB/Fe were investigated using Fourier transform-infrared spectroscopy (FT-IR), X-ray diffraction (XRD), and scanning electron microscopy (SEM). The effect of main parameters such as the adsorbent dosages, pH of the solutions, contact times, and ion concentrations on the efficiency of nitrate removal was investigated. The validity of the experimental method was examined by the isothermal adsorption and kinetic adsorption models. The nitrate sorption kinetics were found to follow the pseudo-second-order model, with a higher determination coefficient (0.99) than the pseudo-first-order (0.86). The results showed that the maximum percentage of nitrate adsorption was achieved at equilibrium pH 5–7, after 120 min of contact time, and with an adsorbent dose of 2 g L^−1^. The highest nitrate adsorption capacity of the modified adsorbent was 19.88 mg g^−1^.

## 1. Introduction

With rapid population growth and economic development, a large amount of nitrogen (N) has been released into surface water and groundwater in a variety of ways, such as via dairy farms and agricultural, industrial, and domestic wastewater discharge [1,2,3,4,5]. Nitrate pollution, which can be hazardous to the environment and human health at high concentrations, has recently received a lot of attention [6,7]. The public is already aware that too much nitrogen in waterways is harmful to aquatic life and that drinking nitrate-nitrogen can result in blue-baby syndrome (methemoglobinemia), so there is an urgent need for a product/technology to solve this problem [8,9]. 

The use of biochar derived from agricultural wastes for the recovery of excess nutrients in aqueous solutions such as contaminated lakes, as well as the use of nutrient-enriched biochar as a soil amendment, can provide a strong solution to a variety of environmental issues. Many researchers recently conducted experiments to test various biological and physicochemical methods of nitrate removal from aqueous solutions, including ion exchange, electrodialysis, catalytic denitrification, and adsorption [3,10,11,12]. However, due to certain restrictions such as the high costs of machines, electric equipment, and operating costs, the secondary pollution involved, and related issues of energy consumption, they cannot be implemented on a large scale [13,14]. The physical and chemical properties of different anions in water have affected recent bioengineering techniques such as ultrafiltration and other membrane processes [15,16]. Physical adsorption by environmentally friendly adsorbents is regarded as one of the safest and most effective methods of removing pollutants from water [17]. Biochar, a stable and porous solid carbon material created by anaerobic biomass pyrolysis, is a favorable material for use in wastewater treatment since it is cost-effective, easy to access, and efficient [16,17,18]. Biochar has a three-dimensional reticulated and porous structure for contaminant adhesion. It is also fast, efficient, and environmentally friendly [7,19,20].

Biochar has attracted much attention due to its excellent adsorption capacity for pollutants in aqueous solution. However, the need for centrifugation and filtration steps may limit the large-scale application of biochar in wastewater [21,22,23]. Biochar surfaces are normally negatively charged and may facilitate positive-charged adsorption, but may be inefficient in adsorption for anions such as nitrate, which are abundant in farm wastewater and priority pollutants for water quality control. As a result, in general, the use of biochar for water quality (nitrate) protection is ineffective and unsustainable [24,25]. 

A range of bioengineering techniques have been introduced to load cations into biochar materials to improve the biochar adsorption capacity for anions. Many studies have demonstrated that Fe-impregnated biochar (Fe/B) is a feasible adsorbent that can effectively remove nutrients from the aquatic solution. Iron is plentiful in the Earth’s crust and environmentally friendly, and Fe/B production is low cost and favorable for industrial production [15,16,26]. Adsorption is recognized for its low price and easy maintenance as the most promising water treatment technology [27]. There is, however, a lack of suitable adsorbents that can be easily separated with enough adsorption capacity and regeneration for further use.

Globally, more than 11 million tons of fresh mushrooms are produced each year from agro-industrial waste [28]. Spent mushroom compost (SMC) is the leftover waste from the mushroom crop after it has been harvested [28,29]. SMC is high in organic nutrients because it is made from wheaten straw, poultry manure, gypsum mixed with cottonseed, and mushroom waste. As a result, its composition, abundance, and low cost have sparked considerable interest in its use as biomass for biochar production [30]. However, SMC is a biomaterial with low density and poor mechanical strength, which limits its application value. Carbonization, possibly of relevance here, has been proposed as a method for improving biomaterial properties [31].

The objective of this research was to determine the sorption effectiveness and the mechanism of pre-treated biochar for the removal of nitrate (NO_3_^−^) in aqueous solution. The physicochemical properties of modified biochar were studied using XRD, FT-IR, and SEM technology. The effects of various parameters such as pH solution, adsorbent dosage, reaction time, initial nitrate concentration, and co-existing anions were investigated on nitrate removal. In addition, the equilibrium and kinetic data of the kinetic model of nitrate adsorption, as well as the adsorption isotherm models, were evaluated.

## 2. Materials and Methods

### 2.1. Preparation of Biochar

Locally available SMC (Meadow Mushrooms Company, Christchurch, New Zealand) was used in this study. The material was washed with deionized water, dried, and ground into small particles no larger than 2 mm in size. Iron chloride hexahydrate (FeCl_3_, 6H_2_O, Merck, Darmstadt, Germany) with analytical grade and without further purification was used to make a 1 molar iron solution. The materials were impregnated with the prepared solution at a solid-to-liquid ratio of 1:10 g mL^−1^. The suspension was stirred for 6 h at room temperature. The mixture was filtered and dried for 48 h at 80 °C. The pristine and pre-treated materials were pyrolyzed in an oxygen-free furnace (Lindberg/Blue M™ 1200 °C Split-Hinge Tube Furnaces, Thermo Fisher Scientific, Waltham, MA, USA) with a nitrogen atmosphere of 150 mL min^−1^, at 600 °C, and at a heating rate of 2 °C/min, then held at the peak temperature for 120 min (Figure 1). The collected biochar SMCB/Fe was rinsed several times with deionized water, dried at 80 °C for 12 h, and crushed to a particle size of less than 0.2 mm. 

### 2.2. Point of Zero Charge

The solid addition method was used to determine the point of zero charge (pHPZC) of biochar [7,32,33]. The initial pH values (pHi) of 0.1 mol L^−1^ of NaCl solutions were adjusted between 2 and 12 using 0.1 mol L^−1^ HCl and 0.1 mol L^−1^ NaOH. Following that, 0.1 g of biochar and 25 mL of NaCl solution were mixed in a 50 mL conical flask and shaken at 25 °C for 48 h. The final pH values (pHf) of the solutions were determined.

### 2.3. Nitrate Desorption Study (Leaching Test)

The leaching test was performed by releasing nitrate ions from SMCB/Fe biochar at various pH levels. Adsorbent (0.1 g) was mixed with 50 mL of distilled water (in five tubes) and the pH levels were adjusted to 1, 4, 5, 7, and 10. The solutions were shaken at room temperature for 48 h, and the supernatant was separated with the help of an external magnet and filtered (0.45 µm membrane) before analysis. The nitrate concentration was determined using ion chromatography (DIONEX ICS-2100, Sunnyvale, CA, USA). Microwave plasma-atomic emission spectroscopy was used to examine the dissolved iron levels (Agilent 4210 MP-AES, Santa Clara, CA, USA). The Bruker ALPHA spectrometer (Bremen, Germany) was used to identify the functional groups of the pristine and pre-treated materials. 

### 2.4. Impact of Co-Adsorption (Competing) Anion

The nitrate adsorption capacity of modified biochar was investigated in the presence of co-existing (competing) anions (e.g., phosphate, sulphate, and chloride). KH_2_PO_4_, Na_2_SO_4_, and NaCl were used to prepare phosphate, sulphate, and chloride solutions, respectively. Experiments were carried out by varying the concentrations of co-existing anions (10, 50, 100, and 200 mg L^−1^) in a nitrate solution (fixed nitrate concentration of 60 mg L^−1^) with an optimum pH and optimal adsorbent dosage of 0.1 g mL^−1^. The mixtures were shaken at room temperature for 120 min at a speed of 120 rpm.

### 2.5. Characterization of Adsorbent

The presence of functional groups on the surface of pristine spent mushroom compost (SMC) and modified spent mushroom compost biochar (SMCB/Fe) before and after adsorption was investigated using FT-IR spectroscopy. The FT-IR spectra ranged from 450 cm^−1^ to 4000 cm^−1^. SEM was used to examine the surface morphology of the Biochar (VP-SEM JSM-IT300; JEOL, Akishima, Japan). The crystalline structure of the pristine and pre-treated materials was investigated using a Bruker X-ray diffractometer (Bremen, Germany) and CuK radiation (=1.5418) in the (2 theta) range of 10 to 90°. 

### 2.6. Adsorption Kinetic

Nitrate solution was made with potassium nitrate. The NO_3_^−^-N concentration in the corresponding stock solution was 1000 mg L^−1^. The subsequent adsorption solution was diluted in accordance with the stock solutions. The solutions were adjusted to pH 6.5 using 0.1 M HCl and agitated on a thermostatic shaker at 150 r min^−1^, at 25 °C. Individual flasks were sampled after specific time intervals (0.03, 0.6, 0.13, 0.16, 0.5, 1, 1.5, 2, 4, 6, 8, 10, 12, 24, 48 h) and filtered through a 0.45 μm membrane (Millipore). The concentration of NO_3_^−^ in the filtrate was determined. Control experiments were carried out on adsorption solutions containing no biochar and those containing a biochar sample mixed with water. All experiments were performed in triplicate, and the average values were calculated. 

The adsorption efficiency (Y) and equilibrium adsorption capacity ($q_{e}$) were calculated using Equations:$$Y=\left(\frac{C_{0}-C_{e}}{C_{0}}\right)\times 100$$
$$q_{e}=\left(\frac{V}{m}\right)\times \left(C_{0}-C_{e}\right)\;$$
where Y is the nitrate adsorption percentage, $C_{0}$ is the initial concentration of target compounds before adsorption (mg L^−1^), $C_{e}$ is the nitrate concentration after adsorption (mg L^−1^), V is the aqueous solution volume (L), and m is the adsorbent dosage (g).

### 2.7. Sorption Isotherm Models

Experiments with various nitrate concentrations (2, 5, 10, 20, 30, 40, 50, 60, 70, 80, 90, and 100 mg L^−1^) were carried out for isotherm studies. The experiments were carried out at 2g L^−1^ biochar, at an initial pH of 6.5, 25 °C, and a shaking rate of 150 r/min for 120 min. The isothermal adsorption process was studied using the Langmuir, Freundlich, and Dubinin–Radushkevich isotherms.

## 3. Results

### 3.1. FT-IR Spectroscopy

Figure 2A depicts the IR spectra of pristine spent mushroom compost with various transmittance peaks for OH stretching vibrations (at 3419 cm^−1^), C-H stretching vibrations (at 2929 cm^−1^), C=O stretching vibrations (at 1690 cm^−1^), aromatic C=C and C-C (at 1501 cm^−1^), and C-O/C-O-H stretching oxygen-containing functional groups (at 1202 and 1049 cm^−1^). The modified mushroom compost (SMCB/Fe) (B) executed all the peaks of oxygen groups, although the intensity decreased as compared to pristine materials (SMC). The magnetic Fe-O bond is identified by a weak stretching peak at 540 cm^−1^. As a result of nitrate uptake over the bio-sorbent (C), the carbonyl and hydroxyl peaks shift to 1675 cm^−1^ and 994 cm^−1^, respectively, due to nitrate (NO_3_^−^) attaching to the carbonyl group (O=C-O...NO_3_^−^) and nitrate linked oxygenate group (C-O...NO_3_^−^) [34].

### 3.2. X-ray Diffractometer (XRD)

The crystallinity of SMC, SMC/Fe, and SMCB/Fe before and after adsorption was investigated using the XRD technique, as shown in Figure 3. The composite XRD patterns include various diffraction signals at 2 thetas of 12, 15, 20, 22, 25, 29, 30, 32, 35, 38, 40, 46, 50, and 51°. These signals indicated that the SMC was well-crystalline. The broad peak in the 2-theta range of 18–26° corresponds to an amorphous compound. When compared to pristine compost (SMC), the modified compost (SMC/Fe) showed fewer signals (12°, 20°, 22°, 25°, 29°, 30°, and 32°) and the intensity of the signals was reduced (crystallinity decreased and amorphous properties increased). Figure 2B depicts the amorphous structure because most of the signals were lost and the intensity of the remaining signals decreased rapidly. Due to the crystalline structure of iron oxide nanoparticles, new signals were observed in C when the XRD patterns of B and C were compared. The XRD profile for SMCB/Fe (before/after) was found to have both an amorphous nature and crystalline structure (amorphous) due to the presence of various diffractions at 2 thetas of 18–26° at 25°, 28°, 30°, 32°, 35°, 36°, 52°, and 66°. Figure 2C shows that the XRD signals are caused by the presence of crystalline magnetic biochar on the surface of biochar. As a result, the XRD profiles of SMCB/Fe (before and after adsorption) showed a nearly identical pattern, indicating that the nitrate had no effect on the crystallinity of the material. 

### 3.3. Surface Morphology of Biochar

The morphology of the biochar was examined using scanning electron microscopy (SEM). Figure 4A shows pristine biochar with a relatively smooth surface that retained some of the intrinsic nature of the raw biomass, which is consistent with the relatively small surface area. The modified biochars had rough and porous surface morphologies. Nanosized flakes were uniformly distributed on the SMCB/Fe surface. The SEM image of SMCB/Fe (before nitrate adsorption) shows that the modified biochar had a porous structure (Figure 4B), indicating that it was composed of a carbonaceous skeleton. The carbonaceous skeleton was broken and pores were blocked after nitrate adsorption (Figure 4C).

### 3.4. Point of Zero Charge (pHpzc)

The pH at which the adsorbent’s net surface charge is zero is known as the pH_pzc_, and at this point, the cation and anion exchange capacities on the adsorbent’s surface are equal [35]. The pH_pzc_ of biochar in the present study was 12, as shown in Figure 5. When the pH of the solution was lower than the pH_pzc_, the net surface charge of modified biochar was positive due to excess H^+^ adsorption. In this case, modified biochar had a high capacity for anionic species adsorption. Furthermore, when the pH of the solution exceeded the pH_pzc_, the net surface charge of surface modified biochar was negative due to H^+^ desorption [7,36]. In this case, the biochar surface was suitable for cation desorption.

### 3.5. Iron and Nitrate Leaching Test

The chemical stability of newly pre-treated biochar was investigated using iron and nitrate desorption (leaching) at various pHs (1–10). The use of both nitrate ion chromatography and microwave plasma-atomic emission spectroscopy failed to detect nitrate and iron in solution. The findings suggest that the proposed biochar can be used to treat water and wastewater without causing secondary pollution in the environment.

### 3.6. Effect of pH on Adsorption Process

The effect of solution pH on nitrate ion adsorption is depicted in Figure 6A. The low adsorption efficiency at high pH values may be attributed to increased competition for adsorption sites by OH^−^ and nitrate ions [7,37]. The electrostatic interaction between the analyte and the protonated adsorbent is most likely responsible for the high nitrate ion adsorption efficiency at pH 2–6. The optimal pH value was between 5–7; as a result, there was no need to adjust the pH of working or real sample solutions.

### 3.7. Effect of Contact Time 

The effect of contact time on nitrate adsorption capacity by SMCB/Fe was investigated using an initial nitrate concentration of 60 mg L^−1^ and adsorbent dosage of 0.1 g mL^−1^ (Figure 6B). The adsorption capacity of SMCB/Fe for removing nitrate was rapid in the early stages of contact time. The rate of adsorption decreased with time after 120 min, and the adsorption process reached an equilibrium within 2 h. This phenomenon could be explained by the presence of a greater number of active sites for nitrate ion adsorption on modified biochar during the early stages of the process. Similar findings were reported by other researchers; they reported that nitrate adsorption increased with contact time and remained constant after reaching equilibrium [34,38,39]. Based on the findings, a contact time of 120 min was selected for future experiments.

### 3.8. Effect of Initial Nitrate Concentration

The effect of initial nitrate concentrations on SMCB/Fe adsorption capacity was investigated by varying the initial nitrate concentrations from 2 to 100 mg L^−1^ (Figure 6C). The results showed that increasing the initial nitrate concentration from 2 to 80 mg L^−1^ increased the adsorption capacity of modified biochar from 0.1 mg g^−1^ to 19.37 mg g^−1^, and increasing the initial nitrate concentration (more than 80 mg L^−1^) had no effect on the adsorption capacity of modified biochar for nitrate. This phenomenon can be attributed to the limited number of active adsorption sites for nitrate adsorption on modified biochar. Other researchers also reported similar results, in which adsorption capacity of adsorbents increased with increasing initial concentrations of nitrate [36,39].

### 3.9. Effect of Co-Existing (Competing) Anions

The effect of co-adsorption anions on nitrate removal by SMCB/Fe was studied with various co-existing anion concentrations (10, 50, 100, and 200 mg L^−1^) in a nitrate solution with a constant nitrate concentration of 60 mg L^−1^ and adsorbent dosage of 0.1 g L^−1^, and reaction time of 120 min at room temperature (22 °C) (Figure 6D). The results showed that the presence of coexisting phosphate, sulphate, and chloride in the water reduced the adsorption capacity of SMCB/Fe toward nitrate. Phosphate and chloride ions had the maximum and minimum effects on nitrate adsorption, respectively. This could be explained by the fact that in the multi-element solution, the electrostatic interaction of coexisting anions with modified biochar adsorption sites was much stronger than that of nitrate ion species. It has been reported that a multivalent anion with a higher charge density is more easily adsorbed than a monovalent anion. Other researchers have reported similar findings, with multivalent and monovalent anions exhibiting more and less adsorption [34,40,41].

### 3.10. Adsorption Kinetic Study

The adsorption of nitrate on SMCB/Fe was significantly fast, with the adsorption equilibrium reached in only 2 h. The pseudo-first-order model was used to represent reversible physical adsorption, whereas the pseudo-second-order model represented chemisorption between the adsorbent and the adsorbate. To validate the rate-controlling step during adsorption, the intraparticle diffusion model was applied [37,42]. In Figure 7 and Figure 8, the kinetic adsorption curve of biochar is depicted. The best-fit parameters of each model and the linear equations for each model are shown in Table 1, where $q_{e}$ and $q_{t}$ (mg g^−1^) are the amounts of NO_3_^−^ adsorbed by the biochar at the equilibrium time and at time *t*, respectively; $k_{1}$(h^−1^), $k_{2}$(g mg^−1^ h^−1^), and $k_{3}$(mg g^−1^ h^−0.5^) are the rate constants of the corresponding model; and C is an intraparticle diffusion constant. 

According to Table 1 and Figure 7b, the nitrate sorption kinetics followed the pseudo-second-order model as they had a higher determination coefficient $R^{2}$ (0.99) than the pseudo-first-order $R^{2}$ (0.86) (Figure 7a). Figure 8a depicts two adsorption steps: the first (Figure 8b) with a sharp slope, indicating that adsorption occurs quickly, possibly due to surface adsorption; and the second (Figure 8c) with a moderate slope, indicating slower adsorption, usually due to intraparticle diffusion. These findings were consistent with a previous research report by Xue et al. [43] showing that the pseudo-second-order model fitted with the nitrate adsorption kinetics data on the biochar/Mg-Fe-layered double hydroxide composite, with a maximum adsorption capacity of 24.8 mg g^−1^, and that the Langmuir model accurately described the adsorption isotherm. 

### 3.11. Adsorption Isotherm Study

Adsorption isotherms depict the relationship between the amount adsorbed by a unit weight of adsorbent and the distribution of adsorbable solute between the liquid and solid phases at different equilibrium concentrations and at a constant temperature. The Langmuir adsorption isotherm model assumes that the surface containing the adsorption sites is homogeneous and that each site can only hold one molecule in thickness (monolayer adsorption) [46]. Heterogeneous surfaces are the main assumptions of the Freundlich adsorption isotherm model [40,46]. The Dubinin–Raduskovich adsorption isotherm model is predicated on the assumption that the characteristic adsorption curve is related to the adsorbent’s porous structure [47].

This model can be used to determine the physical or chemical nature of the adsorption process [39,48]. In Figure 9 and Table 2, the outcomes of Langmuir, Freundlich, and Dubinin–Raduskovich isotherms are shown for nitrate adsorption by modified biochar, where $q_{e},\left({\mathrm{mg}\;g}^{-1}\right)$ is the equilibrium adsorption capacity, $q_{m},\left({\mathrm{mg}\;g}^{-1}\right)$ is the maximum sorption capacity, $C_{e},\left({\mathrm{mg}\;g}^{-1}\right)$ is the residual concentration in solution,$\;k_{L}$ is the Langmuir constant, $R_{L}$ indicates the type of isotherm (either unfavorable ($R_{L}$ > 1), linear ($R_{L}\;$ = 1), favorable (0 < $R_{L}$ < 1), or irreversible ($R_{L}$ = 0)), and $C_{0}$ is the initial concentration of sorbate (mg L^−1^). 

The $k_{f}\left[\left({\mathrm{mg}\;g}^{-1}\right){\left({L\;\mathrm{mg}}^{-1}\right)}^{\frac{1}{n}}\right]\;$ Freundlich constants that correspond to sorption of nitrate and $\frac{1}{n}$ represent the intensity of the adsorption. The $q_{s},\left({\mathrm{mg}\;g}^{-1}\right)$ is Dubinin’s theoretical sorption capacity, $K_{ad}$ is Dubinin’s constant that expresses the energy of the sorption process, ε is the isotherm constant, R is the universal gas constant (8.314 J mol^−1^ K^−1^), T is the temperature (Kelvin), and E equals free energy (kJ mol^−1^). In this study, the Langmuir model with the highest coefficient of determination $R^{2}$ (0.996) provided a better fit for nitrate adsorption than other isotherm models.

As a result, a good fit of the experimental data to the Langmuir model indicated a monolayer nitrate adsorption on a homogeneous surface of SMCB/Fe modified biochar. Other researchers have also reported that Langmuir fitting is useful for nitrate adsorption onto various adsorbents [34,37,43,49,50,51,52,53]. According to the Langmuir isotherm model, the maximum adsorption capacity was calculated to be 19.88 mg g^−1^. The separation factor or equilibrium parameter, $R_{L}$, can be used to predict whether adsorption is favorable. The value of $R_{L}$ indicates the type of the isotherm (either unfavorable ($R_{L}$ > 1), linear ($R_{L}\;$ = 1), favorable (0 < $R_{L}$ < 1), or irreversible ($R_{L}$ = 0)). The $R_{L}$ value obtained in this study was 0.032 (between 0 and 1), indicating that the adsorption isotherm was favorable, according to Table 2. In the Freundlich model, the n value was 4.2 (between 1 and 10), indicating that SMCB/Fe was a favorable nitrate adsorbent.

The nitrate sorption process is confirmed by the adsorption intensity (1 < *n* < 10) in the Freundlich model [54]. Due to unfavorable $R^{2}$ and $q_{s}$ values, the Dubinin–Radushkevich isotherm is unable to predict the sorption pattern for a nitrate ion [34]. As a result, these clarifications point to a monolayer sorption pattern for nitrate. The value of the mean adsorption energy, E, is very useful for predicting the type of adsorption; if it is between 1 and 8 kJ Mol^−1^, the adsorption is physical; if it is between 8 and 16 kJ Mol^−1^, the adsorption is chemical [37,55]. The E value in the free energy model was 1.44, indicating that SMCB/Fe exhibited physical adsorption [25].

### 3.12. Adsorption Mechanism of Nitrates on Biochar

Figure 10 shows that the electrostatic interaction is the most important of the key parameters in the adsorption of nitrate ions using SMCB/Fe. Since nitrate ions are negatively charged and the adsorbent is positively charged over a wide pH range, this can increase the electrostatic interaction between the ions and biochar. The electrostatic interaction between nitrate ions and the positive surface of the adsorbent is demonstrated in Figure 10a. The trapping of nitrate ions in porous biochar is displayed in Figure 10b, and the electrostatic interaction or coordination of nitrate ions and Fe^3+^ cations is showed in Figure 10c.

### 3.13. SMCB/Fe Adsorption Efficacy in Comparison to Other Materials

The salient features of recent studies on the removal of nitrate from aqueous solutions by biochar-supported metal nanoparticles are summarized in Table 3. The nitrate removal capacities of magnetic spent mushroom compost biochar from this study were compared to those of other adsorbents studied by other researchers. These findings indicated that SMCB/Fe had a higher adsorption efficiency than other adsorbents for the removal of nitrate from aqueous solution, such as corn straw, Douglas fir, birch wood, and corncob impregnated with iron metal [5,7,20,34,56,57,58]. However, it has a lower efficiency when compared to magnesium-impregnated soybean and wheat straw. To the best of our knowledge, there is no report in the literature on the application of magnetic spent mushroom compost biochar for nitrate removal, and only a few papers have been published on the biosorption of heavy metals by neat spent mushroom compost. This suggests that the produced modified spent mushroom compost (SMCB/Fe) has the potential to be a suitable adsorbent for nitrate adsorption from aqueous media, and that other metals, such as magnesium, could be used for modification.

## 4. Conclusions and Recommendations for Future Studies

Using spent mushroom compost biochar for nitrate adsorption is a new and innovative polishing process that can be employed for the adsorption of nitrate from the aqueous solution. The results of this study showed that increasing the initial nitrate concentration from 2 to 80 mg L^−1^ increased the adsorption capacity of modified biochar from 0.1 mg g^−1^ to 19.37 mg g^−1^, and increasing the initial nitrate concentration (more than 80 mg L^−1^) had no effect on the adsorption capacity of the modified biochar for nitrate.

The presence of coexisting anions (phosphate, sulphate, and chloride) in the water reduced the adsorption capacity of SMCB/Fe for nitrate. Phosphate and chloride ions had the maximum and minimum effects on nitrate adsorption, respectively.

The data collected were well-adapted to Langmuir isotherms and kinetic models in the pseudo-second-order. Furthermore, the Langmuir isotherm and free energy were proposed as monolayer patterns and physisorption mechanisms for the adsorption process, respectively.

Information on the SMCB/Fe adsorption procedure is needed for environmental policy decisions regarding nitrate clean-up based on robust, scientific evidence with a high degree of accuracy. It is recommended that further research be undertaken in the following areas of interest: SMCB/Fe byproducts such as bio-oil can be used for biofuel production. Yet, there has been little published information on the removal of organic contaminants (e.g., dyes, pesticides) by SMCB/Fe, implying that additional research may need to be conducted. To date, few studies have investigated field trials or applications of magnetic bio-sorbents. These further efforts are pre-requisites for practical applications and commercialization of magnetic bio-sorbents.

## Figures and Tables

**Figure 1 toxics-09-00277-f001:**
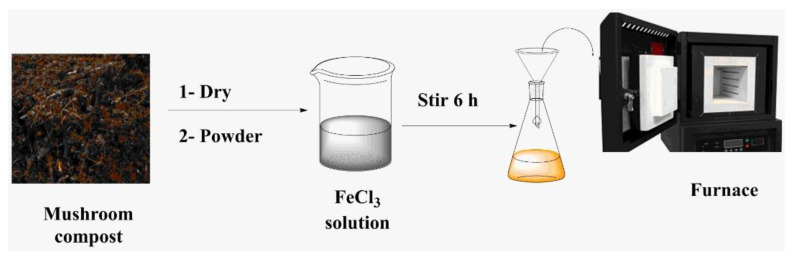
Modification process of SMCB/Fe.

**Figure 2 toxics-09-00277-f002:**
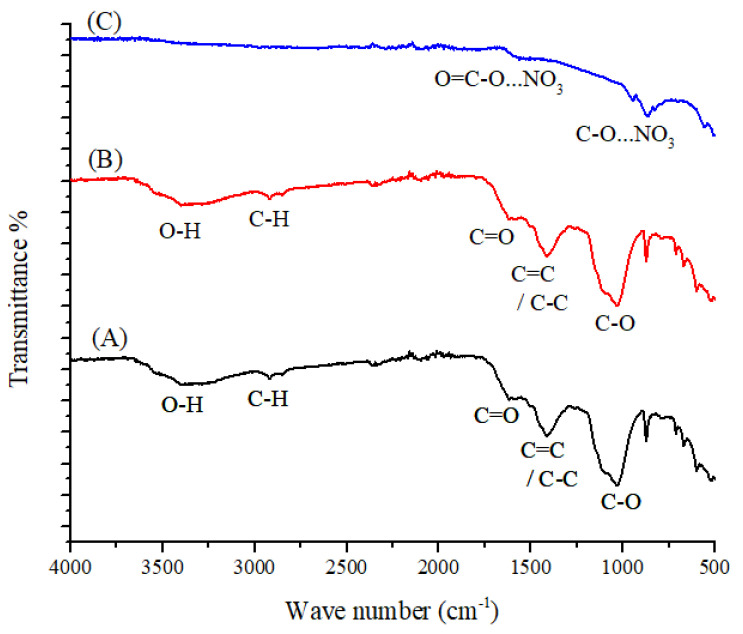
FT-IR spectra of (**A**) SMC, (**B**) SMCB/Fe before nitrate adsorption, and (**C**) SMCB/Fe after nitrate adsorption.

**Figure 3 toxics-09-00277-f003:**
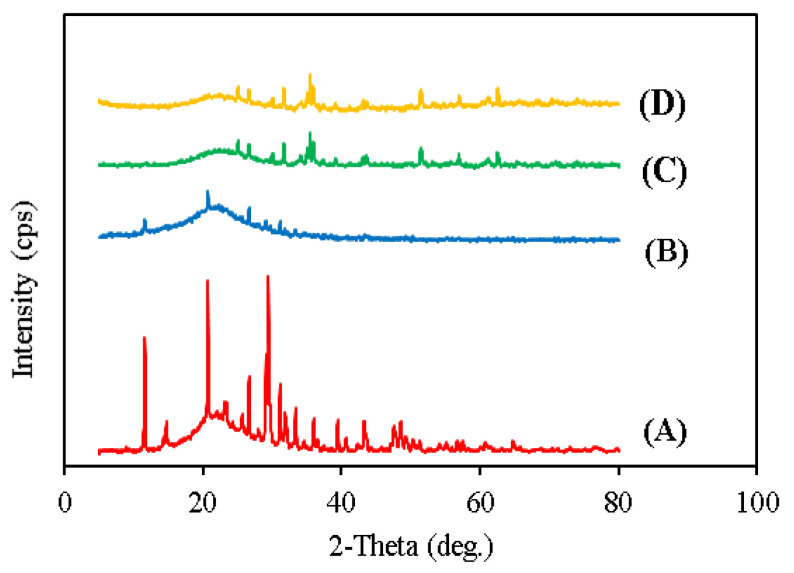
XRD pattern for SMC (**A**), SMC/Fe (**B**), SMC/Fe before adsorption (**C**), and SMCB/Fe after adsorption (**D**).

**Figure 4 toxics-09-00277-f004:**
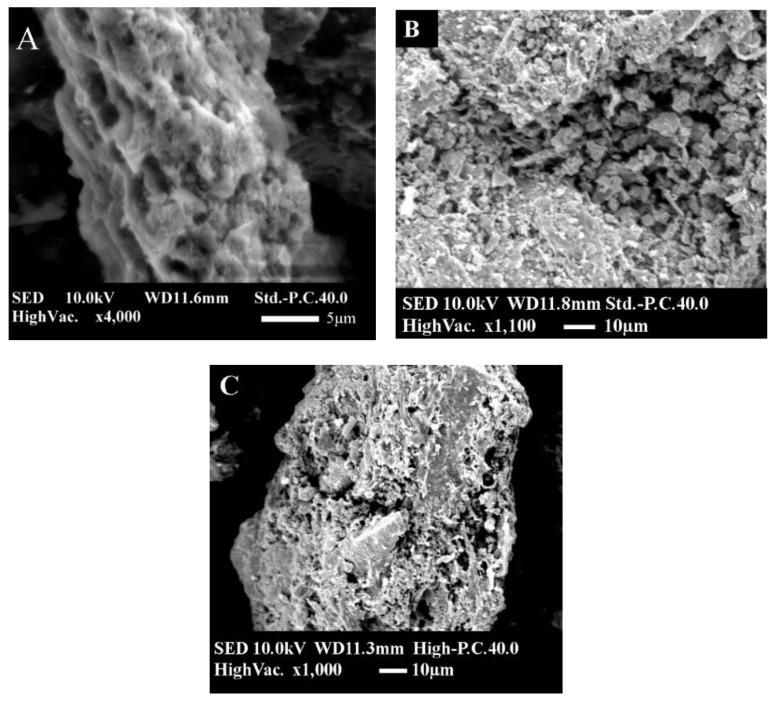
SEM images of SMC. (**A**) Pristine compost, (**B**) SMCB/Fe before nitrate adsorption, and (**C**) SMCB/Fe after nitrate adsorption.

**Figure 5 toxics-09-00277-f005:**
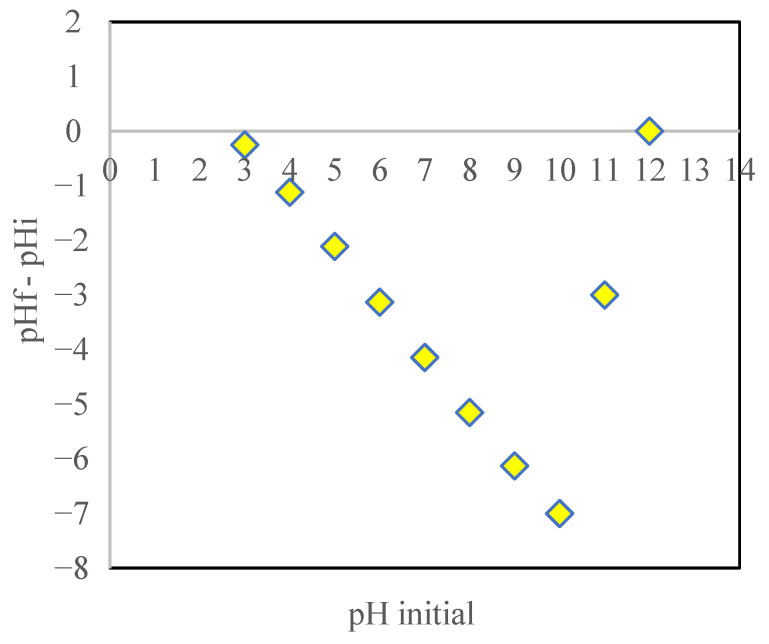
pHpzc of SMCB/Fe.

**Figure 6 toxics-09-00277-f006:**
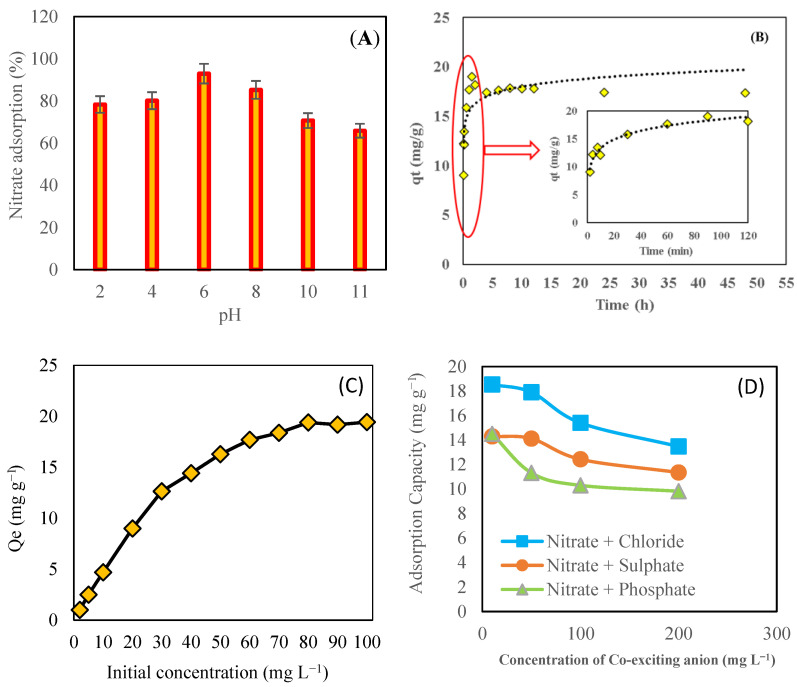
(**A**) Effect of solution pH, (**B**) effect of contact time, (**C**) effect of initial nitrate concentration, and (**D**) effect of co-existing anions on adsorption capacity SMCB/Fe.

**Figure 7 toxics-09-00277-f007:**
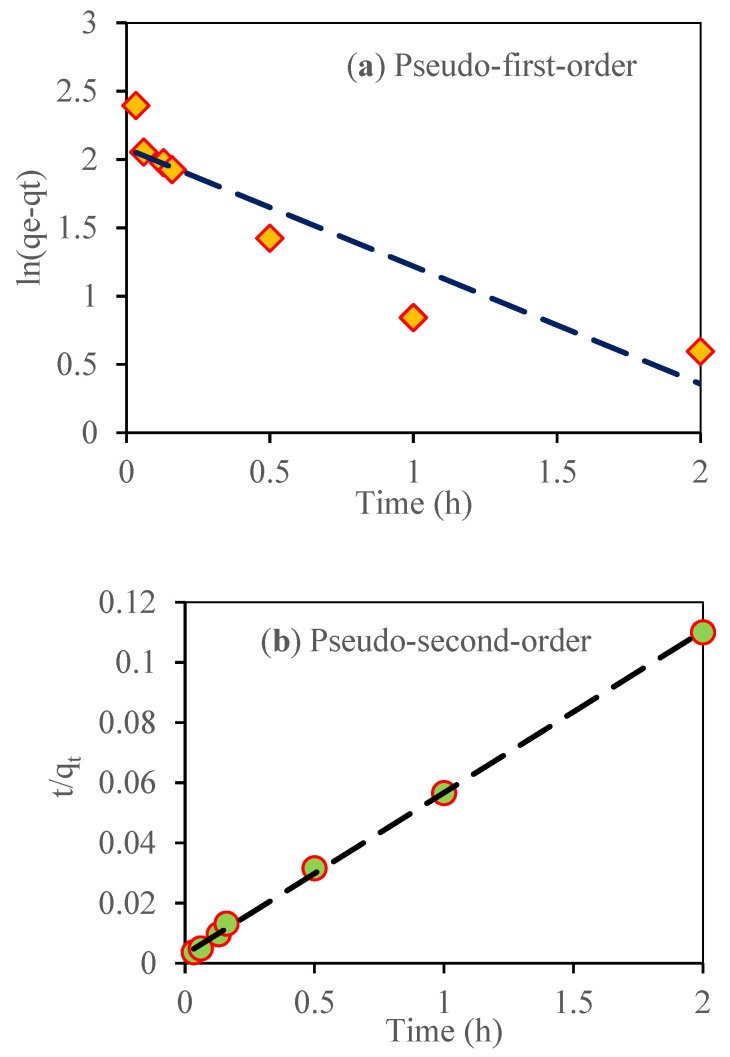
Adsorption kinetics (pseudo-first-order and second-order) of nitrate (initial concentration of nitrate was 60 mg L^−1^). (**a**) Pseudo-first-order; (**b**) Pseudo-second-order.

**Figure 8 toxics-09-00277-f008:**
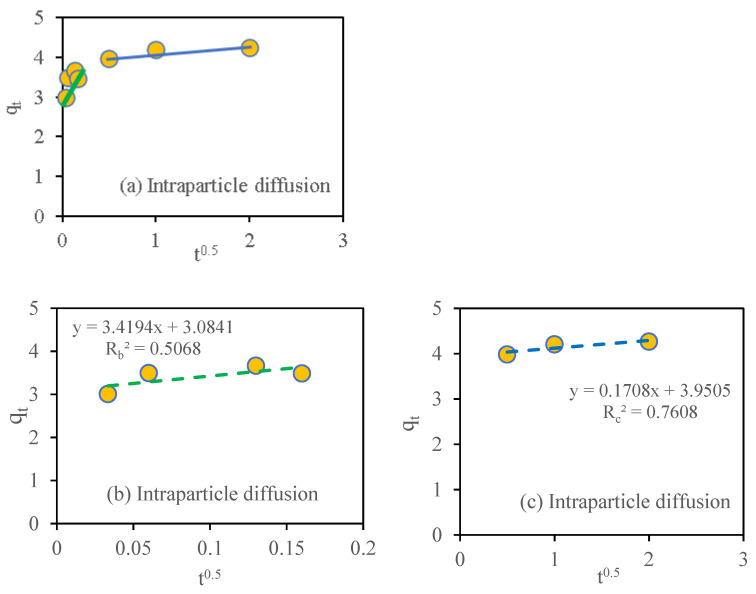
Adsorption kinetics (intraparticle diffusion) of nitrate (initial concentration of nitrate was 60 mg L^−1^). (**a**) Intraparticle diffusion; (**b**) Intraparticle diffusion slope 1; (**c**) Intraparticle diffusion slope 2.

**Figure 9 toxics-09-00277-f009:**
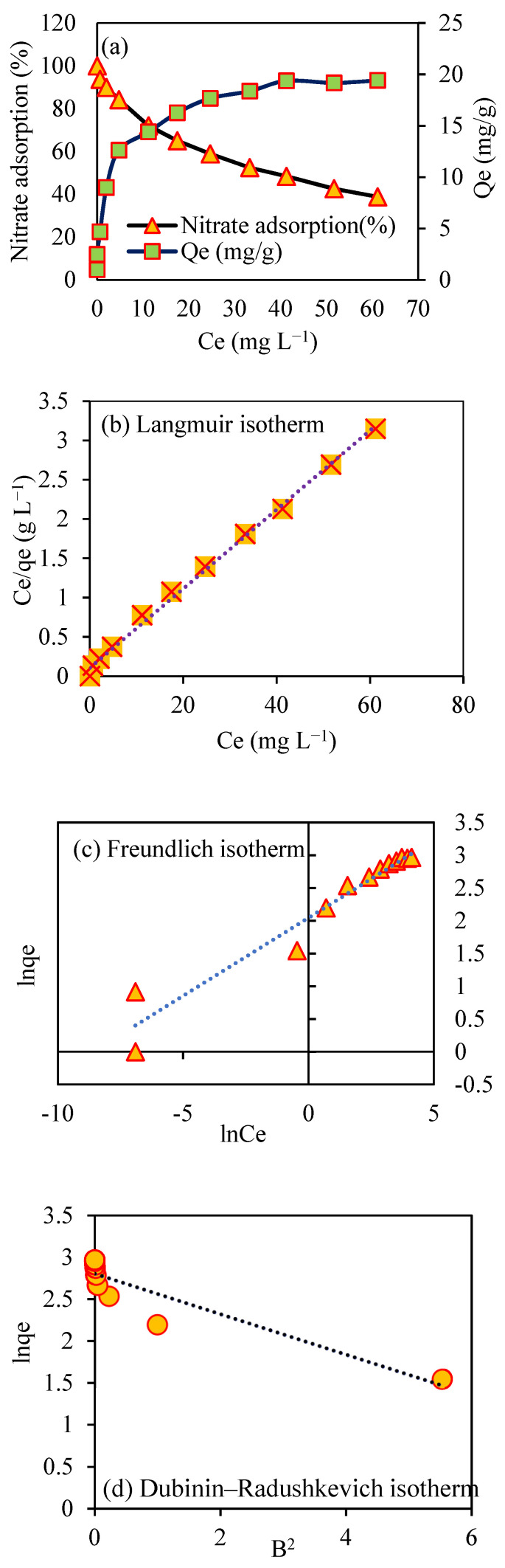
Effect of initial concentration of nitrate on (**a**) adsorption and the adsorption isotherm models of (**b**) Langmuir, (**c**) Freundlich, and (**d**) Dubinin–Radushkevich.

**Figure 10 toxics-09-00277-f010:**
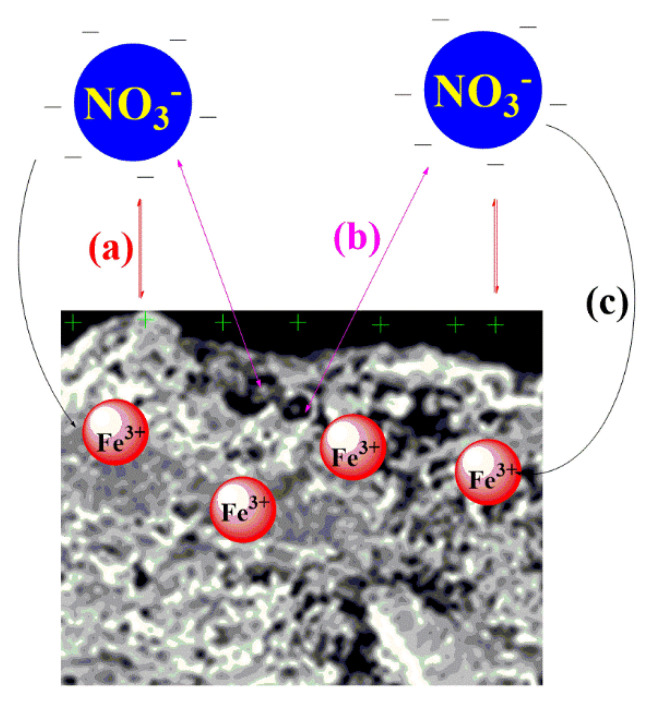
Adsorption mechanism proposed for nitrate ion uptake using SMCB/Fe. (**a**): positive surface of the adsorbent; (**b**): nitrate ions in porous biochar; (**c**): electrostatic interaction or coordination of nitrate ions and Fe^3+^ cations.

**Table 1 toxics-09-00277-t001:** Kinetic parameters of nitrate adsorption by SMCB/Fe.

Model	Reference	Linear Equation	Parameters	Nitrate Adsorption	q_e.exp_ mg g^−1^
Pseudo first order	[31]	$\ln(q_{e}-q_{t})=\ln q_{e}-k_{1}t$	$q_{e}\left({\mathrm{mg}\;g}^{-1}\right)$	20	18.18
			$k_{1}\left(\min^{-1}\right)$	0.861	
			$R^{2}$	0.086	
Pseudo second order	[44]	$\frac{t}{q_{t}}=\frac{1}{k_{2}q_{e}^{2}}+\frac{t}{q_{e}}$	$q_{e}\left({\mathrm{mg}\;g}^{-1}\right)$	18.62	18.18
			$k_{2}\left({\mathrm{mg}\;g}^{-1}\min^{-1}\right)$	0.002	
			$R^{2}$	0.999	
Intraparticle diffusion	[45]	$q_{t}=kt^{0.5}+C$	$k_{b}\;\left(mg\;g^{-1}h^{-0.5}\right)$	3.41	
			$C_{b}$	3.08	
			$R_{b}^{2}$	0.506	
			$k_{c}$	0.17	
			$C_{C}$	3.9	
			$R_{c}^{2}$	0.76	

**Table 2 toxics-09-00277-t002:** Adsorption isotherm models, linear forms, and parameters values for nitrate adsorption using SMCB/Fe.

Model	Reference	Linear Equation	Parameters	Nitrate Adsorption
Langmuir	[37]	$\frac{C_{e}}{q_{e}}=\frac{C_{e}}{q_{m}}+\frac{1}{k_{L}q_{m}}$	$q_{Lm}\left({\mathrm{mg}\;\mathrm{mg}}^{-1}\right)$	19.88
			$k_{L}\left({L\;\mathrm{mg}}^{-1}\right)$	0.503
			$R^{2}$	0.996
			$R_{L}=\frac{1}{1+K_{L}C_{0}}$	0.032
Freundlich	[7]	lnqe=lnqf+(1n)lnCe	$k_{f}\left[\left({\mathrm{mg}\;g}^{-1}\right){\left({L\;\mathrm{mg}}^{-1}\right)}^{1/n}\right]$	6.7
			n	4.2
			$R^{2}$	0.94
Dubinin–Radushkevich	[34]	$\ln q_{e}=\ln q_{s}-k_{ad}\;\left(\varepsilon^{2}\right)\;\;\varepsilon =RT\ln\left[1+\frac{1}{C_{e}}\right]$	$q_{s}\left({\mathrm{mg}\;g}^{-1}\right)$	13.59
			$K_{ad}$	0.024
			$R^{2}$	0.842
Free energy	[37]	E=(2Kad)–12	$E\;\left({\mathrm{kJ}\;\mathrm{mol}}^{-1}\right)$	1.44

**Table 3 toxics-09-00277-t003:** Removal of nitrate contaminants from aqueous solutions by biochar-supported metal nanoparticles.

Biochar Composites	*T* °C	Nitrate (mg L^−1^)	Sorption Isotherm	Kinetic	pH	Max Sorption Capacity (mg g^−1^)	Reference
Corn straw-Fe_2_O_3_	550	50	Langmuir		NA	15.40	[16]
Douglas fir-α-Fe_2_O_3_/Fe_3_O_4_	600	100	Langmuir		2–9	15.00	[56]
Soybean Mg-Al	500	50		*pso	7.5	45.21	[7]
Soybean Mg	500	50		*pso	8.5	77.31	[7]
Birch wood /H_2_O_2_ oxidized	400	500	Langmuir		NA	1.49	[57]
Birch wood /H_2_O_2_ oxidized	600	500			NA	1.09	
Corncob /FeCl_3_	600	2000	Langmuir		3–7	14.46	[20]
Corncob/FeCl_3_	600	100	Langmuir		3–11	32.33	[58]
Sugarcane bagasse/ ECH	600	100	Langmuir	*pso	3	28.21	[34]
Wheat-straw/Mg-Fe	600	45	Langmuir	*pso	7–8	24.8	[5]
Spent mushroom compost/FeCl_3_·6H_2_O	600	60	Langmuir	*pso	5–7	19.88	this study

*pseudo-second-order.

## Data Availability

This study includes all supporting data, which can be obtained from the corresponding authors upon request.

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
