# Peer review of "Application of Modified Spent Mushroom Compost Biochar (SMCB/Fe) for Nitrate Removal from Aqueous Solution"

_toxics, 2021, doi:10.3390/toxics9110277_

Round 1
Reviewer 1 Report
The study is based on the spent mushroom compost biochar (SMCB/Fe) coated with iron chloride hexahydrate. The study evaluates the essential findings of nitrate adsorption using the SMCB/Fe particles. The overall presentation of the article requires significant improvements with a focus on the mode of units representation of the article. Some analysis justifications and nomenclature are inconsistent. The article used biochar in most places; somewhere, SMCB/Fe nanoparticles have been used. However, the author could use the word nanoparticle only if the size of SMCB/Fe reaches (10-9 = nm). The particle size was not mentioned clearly in the article to understand whether the studied material falls under the nanoparticle category or not.
Comments:
- In the abstract, there was a lack of background information on the need for the study. The use of units in the abstract is inconsistent. The author used g.L-1; in this form, -1 should be used in a superscript.
- There is a missing full stop on line number 53.
- Line 54-58 context is not clear. The author should re-explain what they are trying to say here. The biochar has a negatively charged surface that can attract nitrate ions having a negative charge, and how is it possible? It requires some clarification; even if they need to have a surface coating of any other material with a positive charge, it must be briefly explained here.
- Line 66-67, SMC should be abbreviated somewhere in the text before it is used.
- The last paragraph of the introduction should be adjusted in the section on material and methodology. Instead, the objective of the study may be to improve the strength of the article here.
- Line 100-101, all the units in the document need a revision. There is inconsistency in the entire document.
- Line 127 and in abstract, the use of sign for the degree is not recommended; therefore, the authors should modify the expression of the degree.
- Line 133-134, the time interval should be in the same units.
- Line 143-144, the units are inconsistent, equations 1 and 2 are very general, and references must be provided from a similar study. Equations 3, 4, and 5 must also need a reference.
- Line 159-160, Equation 1 is placed again while previously used in the article on page 3.
- On page 4, please follow the instructions from the journal for the display of equations.
- Line 178-191, there is a lack of reference provision; please ensure the source for the estimate about transmittance at several wavenumbers.
- Section 3.2, Line 196-211 requires revision with details about the difference between SMCB and SMCB/Fe. Of course, simple SMCB is a non-crystalline structure, so it could not produce the diffraction angle (2ɵ).
- Figure 5D has a description of “onion” in the caption; please choose the correct word to depict the illustration.
- While mentioning the surface area and active sites, mentioning the surface area (m2/g) when developing a charged surface on a particle is better to estimate the surface area by Brunauer-Emmett-Teller (BET) analysis.
- Line 207: there is no full sentence. The crystalline 207caused by the presence of magnetic FeCl3 nanoparticles on the surface of biochar.
- Line 289 needs a revision; please check the wordings and sentences.
- Line 349-354, “E” notation is used instead of ɛ, which is confusing. The whole paper needs a severe revision on the units, supervised before submitting again after revisions. It should be a serious responsibility of the corresponding author to check the consistency of the paper.
- Dubinin-Radushkevich model showed unfavorable R2 and qs, which is not considered according to the authors to evaluate the sorption of nitrates. However, the same model was used to justify mean adsorption energy (ɛ) sounds unjustified. The reference was mentioned in the results, which could not explain a lot if the Langmuir isotherms are used to justify the monolayer structure to the adsorbent.
- From figure 8D, it is evident that there is no match of the data to justify anything related to sorption hence the model selection must be forfeited from any analysis scientifically, especially for the case of adsorption energy. The analysis must be scientific and logical, which is not used in practice.
- The conclusion needs a significant revision, as it seems to be just a revision of the abstract. The repetition of abstract and conclusion is not recommended. However, the author may revise it with seriousness.
Author Response
Dear reviewer,
Thank you for your insightful comments and suggestions on our manuscript's structure. We have made the required changes to the manuscript. Changes/explanations have been made in response to your comments and are reflected in the revised manuscript's track changes (red colour).

Reviewer 2 Report
The manuscript Application of modified spent mushroom compost biochar (SMCB/Fe) for nitrate removal from aqueous solution is dealing with nitrate removal by biosorbent produced from spent mushroom compost. Since the awareness of nitrate pollution is greater than ever, research on nitrate removal are of great importance.
This manuscript is well organized but there are few remarks:
Line 23 correct unit: mg g-1
Line 45 please rephrase „such as ultrafiltration and membrane removal“. Maybe it would be better to write such as ultrafiltration and other membrane processes.
Line 53 full stop is missing
Lines 147-149 please write the correct kinetic formula
Line 152 it would be great if you could write what for the C constant stand for.
Lines 161-163 please check the equations and decide whether you will write some constant in capital or small letters
I think it would be better to write all equations in the manuscript with „Equation“ function, because this pasted images are not clear enough and the lines are mixed up a little bit.
Line 188 „after uptake of nitrate“
Figure 2 – the “(A)” slipped a little away. Delete the comma in the Figure caption after (A)
Figure 3 – the figures are not marked as “A, B or C” and are very unclear and blurry.
Why do you have used the concentration of 60 mg/L N-NO3- for the experiments?
Lines 282-283 What do you mean by maximal and minimal. It would be good to explain it differently.
Line 289 Delete the last sentence – the subsection title
291 I think this sentence should be deleted since the biosorbent was not chosen only for kinetic studies - it was examined in this study.
Line 299 the form you have used for explaining the units (qe = mg g-1, etc.) are confusing, It would be better to write it like this qe (mg g-1)… Again, it would be better to write math formular via the “Equation” tool.
Kinetics: it should be detailed explained and you haven’t explained the intraparticle diffusion at all.
Table 1 and Figure 7 – why do you have added the 2nd and 3rd picture in Figure 7, when you have not presented the results in Table 1. In addition, the k of intraparticle diffusion in Table 1 and Figure 7 do not match.
Check the text in caption of Table 2.
Lines 360-369 should be added to the text before Figure 8 and Table 2 to make it a whole.
Line 376 references are missing.
Lines 382-384 one part of sentence was duplicated.
Table 3 – it is nicer to write the abbreviation for Temperature as T.
Lines 413-414 Please rephrase this sentence.
Line 420 Delete the last sentence “5. Conclusions”
Please check the manuscript and correct chemical formula and units and write them in super- and subscript accordingly. Uniform capital and small letters for units and symbols.
Author Response
Dear reviewer,
Thank you for your thoughtful comments and suggestions on the structure of our manuscript. We have made the required changes to the manuscript. Changes/explanations have been made in response to your comments, and they are reflected in the revised manuscript's track changes (red colour).

Reviewer 3 Report
Reviewer Report
Manuscript Number:1390707
Title: Application of modified spent mushroom compost biochar (SMCB/Fe) for nitrate removal from aqueous solution
The manuscript deals with obtaining and application of modified spent mushroom compost biochar (SMCB/Fe) for nitrate removal from aqueous solution. The surface properties and morphology of SMCB/Fe were investigated using Fourier transform-infrared spectroscopy, X-ray diffraction, and scanning electron microscopy. The effect of adsorbent dosages, pH-values of the solutions, contact times and nitrate ion concentrations on the adsorption efficiency of nitrate removal, was investigated. The experimental conditions effect on the adsorption efficiency was checked and the kinetic and equilibrium studies were modeled using the PFO, PSO, and intraparticle diffusion models as well as the Langmuir, Freundlich, and Dubinin-Radushkevich ones.
Overall, the subject is of big interest to readers specialized in pollutants removal from water but the presentation of the data and content must be improved. The paper is sloppy and do not correspond to the requirements of the journal TOXICS. Authors should see again the template and requirements for paper preparations. Taking into account the general comments and details I recommend major revision of the paper.
Here I list the arguments that led me to the remarks above mentioned:
Introduction: There is no analysis of works about preparation and application of the proposed adsorbent, so the novelty of the study is not clear.
Line 41: It should be noted what metal salts were used as adsorbents. What are “chemical adsorbents”?
Line 88: It should be check chemical formulas.
Line 112: It should be noted what specific phosphate, sulphate, and chloride salts were used in the study.
Section 2.6- 2.7: It should carefully check all formulas, their numbering, units and values, since the sections are written extremely casually and in some cases information is wrong.
Line 231-236: It is unclear why the charge of the adsorbent is positive in an alkaline environment.
Fig.5 (B): It is necessary to show the initial plot within 120 minutes.
Fig. 6: It should be check axis designations considering the linear form equation of the PSO model.
Fig 7: There is no explanation for the obtained results.
Table 1: There is no qe, exp -value and it is not used in explanation of PFO and PSO models.
Fig.8 : It should be check “Dubinin-radushkevic” and “Dubinin-radushkevich.
The magnetic adsorbent has been obtained, but its magnetic properties have not been quantitatively described.
It should be represented figure about preparation of the adsorbent and possible interactions between the adsorbent and ions metals.
Author Response
Dear reviewer,
Thank you for your mindful comments and suggestions on the structure of our manuscript. The required changes have been made to the manuscript. In response to your comments, changes/explanations have been made and are reflected in the revised manuscript's track changes (red colour).

Reviewer 4 Report
The structure of the manuscript is clear, and the content is well presented. In my opinion this work can be published by Toxics after some revisions.
Line 37. Please supplement with an additional reference. I would like to suggest these:
Raboni, M.; Viotti, P.; Rada, E.C.; Conti, F.; Boni, M.R. The Sensitivity of a Specific Denitrification Rate under the Dissolved Oxygen Pressure. Int. J. Environ. Res. Public Heal. 2020, Vol. 17, Page 9366 2020, 17, 9366, doi:10.3390/IJERPH17249366.
Line 41. Please supplement with an additional reference. I would like to suggest these:
Boni, M.R.; Chiavola, A.; Marzeddu, S. Remediation of Lead-Contaminated Water by Virgin Coniferous Wood Biochar Adsorbent: Batch and Column Application. Water, Air, Soil Pollut. 2020, 231, 171, doi:10.1007/s11270-020-04496-z.
Boni, M.R.; Marzeddu, S.; Tatti, F.; Raboni, M.; Mancini, G.; Luciano, A.; Viotti, P. Experimental and Numerical Study of Biochar Fixed Bed Column for the Adsorption of Arsenic from Aqueous Solutions. Water (Switzerland) 2021, 13, 915, doi:10.3390/w13070915.
Line 53. Please check if a point is missing.
Line 66. Please enter appropriate references.
Line 122, 151, 152. Please check all constant symbols and units of measure in the manuscript.
Line 141, 142, 147, 148, 149. Please check the equations: they are all the same. Insert appropriate references related to the models used.
Line 159-170. Please check all the formatting of the equations.
Line 289, 420. Please check.
Line 319. Please check if two values were exchanged in the table.
As regards the discussion of the results, appropriate bibliographic references should be indicated.
Author Response
Dear reviewer,
Thank you for taking the time to make thoughtful comments and suggestions about the structure of our manuscript. The required adjustments to the manuscript have been made. Changes/explanations have been made in response to your comments, and they are reflected in the revised manuscript's track changes (red colour).

Round 2
Reviewer 1 Report
The authors have addressed all my comments.
Reviewer 3 Report
You can find comments in attached file

Reviewer 4 Report
All comments have been addressed and all the relevant modifications have been made to the manuscript which make it publishable in its present form.